# Dual Monoclonal Antibodies on Sars-Cov-2 Alpha and Delta Variants: Clinical and Virological Efficacy

Valentine Marie Ferré,[a,b] Nathan Peiffer-Smadja,[a,c] Laura Kramer,[d] Romain Coppée,[a] Aïcha Kante,[c] Margaux Debarge,[d] Christophe Choquet,[e] Thibault Saint Joannis,[b] Donia Bouzid,[a,e] Jonathan Messika,[f,g,h] Jennifer Le Grand,[d] Michael Thy,[c,i] Solen Kernéis,[a,j] Diane Descamps,[a,b] Benoit Visseaux,[a,b] Jade Ghosn[a,c]

[a]Université Paris Cité and Université Sorbonne Paris Nord, Inserm, IAME, Paris, France
[b]Virology Department, AP-HP, Bichat-Claude Bernard Hospital, Paris, France
[c]Infectious Disease Department, AP-HP, Bichat-Claude Bernard Hospital, Paris, France
[d]Pharmacy Department, AP-HP, Bichat-Claude Bernard Hospital, Paris, France
[e]Emergency Department, AP-HP, Bichat-Claude Bernard Hospital, Paris, France
[f]Université Paris Cité and Université Sorbonne Paris Nord, INSERM U1152 PHERE, Paris, France
[g]Pneumology Department, AP-HP, Bichat-Claude Bernard Hospital, Paris, France
[h]Paris Transplant Group, Paris, France
[i]Université Paris Cité and Université Sorbonne Paris Nord, EA 7323–Pharmacology and Therapeutic Evaluation in Children and Pregnant Women, Paris, France
[j]Equipe de Prévention du Risque Infectieux (EPRI), AP-HP, Hôpital Bichat, Paris, France

**ABSTRACT** Monoclonal antibodies (MAbs) targeting the Spike glycoprotein of SARS-CoV-2 is a key strategy to prevent severe COVID-19. Here, the efficacy of two monoclonal antibody bitherapies against SARS-CoV-2 was assessed on 92 patients at high risk of severe COVID-19 between March and October 2021 (Bichat-Claude Bernard Hospital, Paris, France). Nine patients died despite appropriate management. From 14 days following treatment initiation, we observed a slower viral load decay for patients treated with the bitherapy Bamlanivimab/Etesevimab compared to the Casirivimab/Imdevimab association therapy ($P = 0.045$). The emergence of several mutations on the Spike protein known to diminish antiviral efficacy was observed from 1 to 3 weeks after infusion. The Q493R mutation was frequently selected, located in a region of joint structural overlap by Bamlanivimab/Etesevimab antibodies. Despite that this study was done on former SARS-CoV-2 variants (Alpha and Delta), the results provide new insights into resistance mechanisms in SARS-CoV-2 antibodies neutralization escape and should be considered for current and novel variants.

**IMPORTANCE** Monoclonal antibody bitherapies (MAbs) are commonly prescribed to treat severe SARS-CoV-2-positive patients, and the rapid growth of resistance mutation emergence is alarming globally. To explore this issue, we conducted both clinical and genomic analyses of SARS-CoV-2 in a series of patients treated in 2021. We first noticed that the two dual therapies prescribed during the study had different kinetics of viral load decay. Rapidly after initiation of the treatments, resistance mutations emerged in the interface between the MAbs and the target Spike glycoprotein, demonstrating the importance to continuously screen the viral genome during treatment course. Taken together, the results highlight that viral mutations may emerge under selective pressure, conferring a putative competitive advantage, and could rapidly spread, as observed for the Omicron variant.

**KEYWORDS** COVID-19, SARS-CoV-2, Bamlanivimab/Etesevimab, Casirivimab/Imdevimab, monoclonal antibodies, mutation selection

Monoclonal antibodies (MAbs) targeting the spike glycoprotein of SARS-CoV-2 is a key option to prevent severe COVID-19. Bamlanivimab monotherapy presented low efficiency and high rates of viral mutation emergence (1) and was quickly replaced by MAbs

Address correspondence to Valentine Marie Ferré, valentinemarie.ferre@aphp.fr.

The authors declare a conflict of interest. V.M.F. reports congress accommodation from Gilead outside the submitted work. J.M. received congress accommodation from Biostest and CSL Behring outside the submitted work. D.D. has received personal fees from Gilead-Sciences, ViiV Healthcare, MSD, Janssen-Cilag, and research grants from Gilead-Sciences and ViiV Healthcare, outside the submitted work. J.G. reports personal fees from Merck, grants and personal fees from ViiV healthcare, grants and personal fees from Gilead Sciences, personal fees from Roche, personal fees from AstraZeneca, personal fees from Janssen, personal fees from TheraThechnologies outside the submitted work. The other authors declare having no conflict of interest with the current work.

dual therapies. Emergency use authorization was provided in March 2021 in France for Bamlanivimab and Etsevimab (B/E) and Casirivimab and Imdevimab (C/I) for patients at high risk of severe COVID-19 within 5 days after the first positive test or symptom onset. Randomized clinical trials demonstrated lower COVID-19-related hospitalizations and deaths in the general population with these dual therapies (2, 3). However, no data on viral sequencing and potential resistance mutation appearance were available. This real-life monocentric cohort study conducted between March and October 2021 in Bichat-Claude Bernard University Hospital (Paris, France) describes the clinical safety and efficacy of B/E and C/I in a population at high risk of severe COVID-19 and depicts the viral load evolution and viral mutation emergence according to each dual therapies. Patients requiring oxygen supplementation at diagnosis were excluded. Both dual therapies were administered up to May 12, 2021, but only C/I was recommended by health authorities due to the Delta variant local emergence. Patients received B/E (700/1,400 mg) or C/I (1,200/1,200 mg) as single intravenous infusion. Clinical follow-up was carried out until discharge for in-patients and every week until viral clearance for outpatients. Virological follow-up included a nasopharyngeal (NP) swab right before infusion (D0), on days three (D3) and seven (D7) postinfusion, and weekly until viral load clearance. All patients provided their oral consent for data collection.

RT-PCR was performed on each respiratory sample, and the whole SARS-CoV-2 genome was sequenced (MinION; Oxford Nanopore).

Overall, 92 patients with a median age of 69 years received MAbs dual therapy. Overall, 55% of the patients had no SARS-CoV-2 vaccination history. Patients' comorbidities and treatments are depicted in the Table 1, and the characteristics were similar between the B/E and C/I treatment groups. The median delay between symptom onset and MAbs infusion was 3 days. Patients' severity assessed by the World Health Organization (WHO) ordinal scale was different across B/E and C/I groups at D0 ($P < 0.0001$, Fisher exact test), with a higher proportion of severe patients in the C/I group, while similar on D7 postinfusion for both arms ($P = 0.39$). Other outcome criteria were similar with both treatments. Fourteen patients died (seven in both groups), including nine deaths directly related to COVID-19 worsening. Among the surviving patients, 69% did not require oxygen therapy (24/31 and 27/43 for B/E and C/I, respectively). The hospitalization duration was significantly different with a median at 15 and 7 days, respectively, for the B/E and C/I groups ($P = 0.009$; Table 1). No severe adverse event related to MAbs occurred during follow-up.

SARS-CoV-2 variant was determined for 89/92 infections with 54% Alpha, 28% Delta, and 14% other variants (Table 1). Patients presented similar initial viral loads at infusion in both groups with a median threshold cycle ($C_T$) value of 18.2 [14.9 to 23.7] and a similar viral decay on D7 postinfusion with 19/26 and 16/24 showing a NP $C_T$ value $< 30$ ($P = 0.75$), respectively. Between D14 and D21 postinfusion, a lower viral decay was observed for B/E versus C/I groups (8/12 versus 3/14 NP with $C_T$ value $<30$; $P = 0.045$) (Fig. 1A). Focusing on patients infected with an Alpha variant ($n = 50$), median $C_T$ value was also similar at D0 and D7 postinfusion. Between D14 and D21 postinfusion, 8/12 had a $C_T$ value $<30$ in the B/E group versus none of the three Alpha-infected patients in the C/I group (Fig. 1B).

When considering the patients receiving a C/I dual therapy and infected by the Alpha ($n = 16$) or Delta ($n = 26$) variant, $C_T$ values were not different at D0, D7, and $>14$ days postinfusion.

Several spike mutation emergences were observed among the 20 patients treated with B/E or C/I with available sequences in follow-up ($n = 12$ and 8 Alpha and Delta variants, respectively). All mutations were detected between 5 and 18 days after infusion (Fig. 1C). We observed a trend for more frequent mutation appearances with B/E bitherapy (7/10) and Alpha variant than with C/I therapy occurring only once on a Delta variant (1/8; $P = 0.03$; Fig. 1D).

In our study, both therapies were well tolerated and associated with similar outcomes regarding the number of deaths and oxygen requirements, but we observed a higher level of death than in the B/E or C/I trials (2, 3), which may be explained by including older or at higher risk patients. The viral decay observed was in line with previous findings (2–4) but a

**TABLE 1** Epidemiological, clinical, and virological data for included patients[a]

| Patient's characteristics | All patients ($n = 92$) | Bamlanivimab/Etesivimab ($n = 40$) | Casirivimab/Imdevimab ($n = 52$) | P value |
|---|---|---|---|---|
| Age (yr) | 69 [57–79] | 71 [59–83] | 67 [55–76] | $P = 0.15$ |
| Sex (female) | 41 (46) | 19 (47) | 22 (42) | $P = 0.62$ |
| **Vaccination state at infusion** | | | | |
| No vaccination | 51 (55) | 27 (68) | 24 (46) | $P = 0.06$ |
| Previous vaccination | 41 (45) | 13 (32) | 28 (54) | |
| 1 dose of vaccine | 17 (18) | 12 (30) | 5 (10) | |
| 2 doses of vaccine | 16 (17) | 1 (2) | 15 (29) | |
| 3 doses of vaccine | 8 (9) | 0 | 8 (15) | |
| **Coexisting conditions** | | | | |
| BMI >30 | 19/65 (29) | 11/29 (38) | 8/36 (22) | $P = 0.18$ |
| Transplantation | 25/92 (27) | 10/40 (25) | 15/52 (29) | $P = 0.81$ |
| Chronic renal insufficiency | 35/92 (38) | 15/40 (38) | 20/52 (38) | $P = 0.96$ |
| Type II diabetes | 41/92 (45) | 16/40 (40) | 25/52 (48) | $P = 0.53$ |
| Chronic heart insufficiency | 18/92 (20) | 8/40 (20) | 10/52 (19) | $P = 1.0$ |
| Chronic respiratory disease | 12/91 (13) | 6/39 (15) | 6/52 (12) | $P = 0.8$ |
| ≥3 comorbidities | 16/92 (17) | 7/40 (18) | 9/52 (17) | $P = 1.0$ |
| **Long-term medication** | | | | |
| Immunosuppressive therapy | 46/92 (50) | 16/40 (40) | 30/52 (58) | $P = 0.07$ |
| Chemotherapy | 11/92 (12) | 3/40 (8) | 8/52 (15) | $P = 0.08$ |
| **Clinical outcomes** | | | | |
| Delay since symptoms onset (days) | 3 [2–4] | 3 [2–4] | 3 [1–4] | $P = 0.45$ |
| Dexamethasone | 20/88 (23) | 8/38 (21) | 12/50 (24) | $P = 0.8$ |
| Tocilizumab | 7/88 (8) | 2/38 (5) | 5/50 (10) | $P = 0.69$ |
| $O_2$ requirement | 34/87 (39) | 12/37 (33) | 22/50 (44) | $P = 0.37$ |
| Hospitalization | 72/92 (78) | 29/40 (73) | 43/52 (83) | $P = 0.3$ |
| Hospitalization duration (days) | 13 [5–19] | 15 [9–28] | 7 [4–16] | $P = 0.009$ |
| ICU | 7/88 (8) | 2/38 (5) | 5/50 (10) | $P = 0.35$ |
| Death related to COVID | 9/92 (10) | 5/40 (13) | 4/52 (8) | $P = 0.5$ |
| **Baseline variant** | | | | $P < 0.001$ |
| Historical | 6 (7) | 3 (8) | 3 (6) | |
| Alpha | 50 (54) | 34 (85) | 16 (30) | |
| Beta | 6 (7) | 2 (5) | 4 (8) | |
| Gamma | 1 (1) | 1 (2) | 0 | |
| Delta | 26 (28) | 0 | 26 (50) | |
| Unknown | 3 (3) | 0 | 3 (6) | |
| **Virological follow-up** | | | | |
| NP $C_T$ values at infusion[b] | 18.2 [14.9–23.7] | 18.0 [14.7–23.5] | 18.2 [15.6–25.0] | $P = 0.6$ |
| $C_T$ <30 at infusion[b] | 79/88 (90) | 36/39 (92) | 43/49 (88) | $P = 0.7$ |
| $C_T$ <30 at D7 after infusion[c] | 31/56 (55) | 19/30 (63) | 12/26 (46) | $P = 0.28$ |
| $C_T$ <30 D14 to D21 after infusion | 8/20 | 8/12 | 0/8 | $P = 0.005$ |
| Patients with viral sequence >D3 | 18 | 10 | 8 | |
| Patients with spike mutation(s) | 8/18 | 7/10 | 1/8 | $P = 0.03$ |

[a]Results are presented as median [interquartile range] for continuous variables and $n$ (%) for qualitative variables. BMI, body mass index; ICU, intenvsive care unit.
[b]From 3 days preinfusion to D1 postinfusion.
[c]D7 ± 48 h.

higher NP viral loads and a more frequent selection of mutations in the viral spike for patients treated with B/E was depicted. The Q493R mutation frequently selected in our study has already been described as selected in a patient infected by an Alpha variant and receiving Bamlanivimab/Etesevimab (5). Interestingly, it is now naturally present for the first time in a widely circulating variant with the Omicron emergence. Structurally, Q493 is not located in the ACE2-binding ridge but in a region of joint structural overlap by B/E antibodies (Fig. 1E). Similarly to the E484K substitution, the introduction of a bulky, positively charged residue instead of glutamine at position 493 likely affects the binding of some antibodies (6) (Fig. 1F). The Q493R mutation has been described as associated with reduction of the 50% inhibitory concentration for both Bamlanivimab and Etesivimab

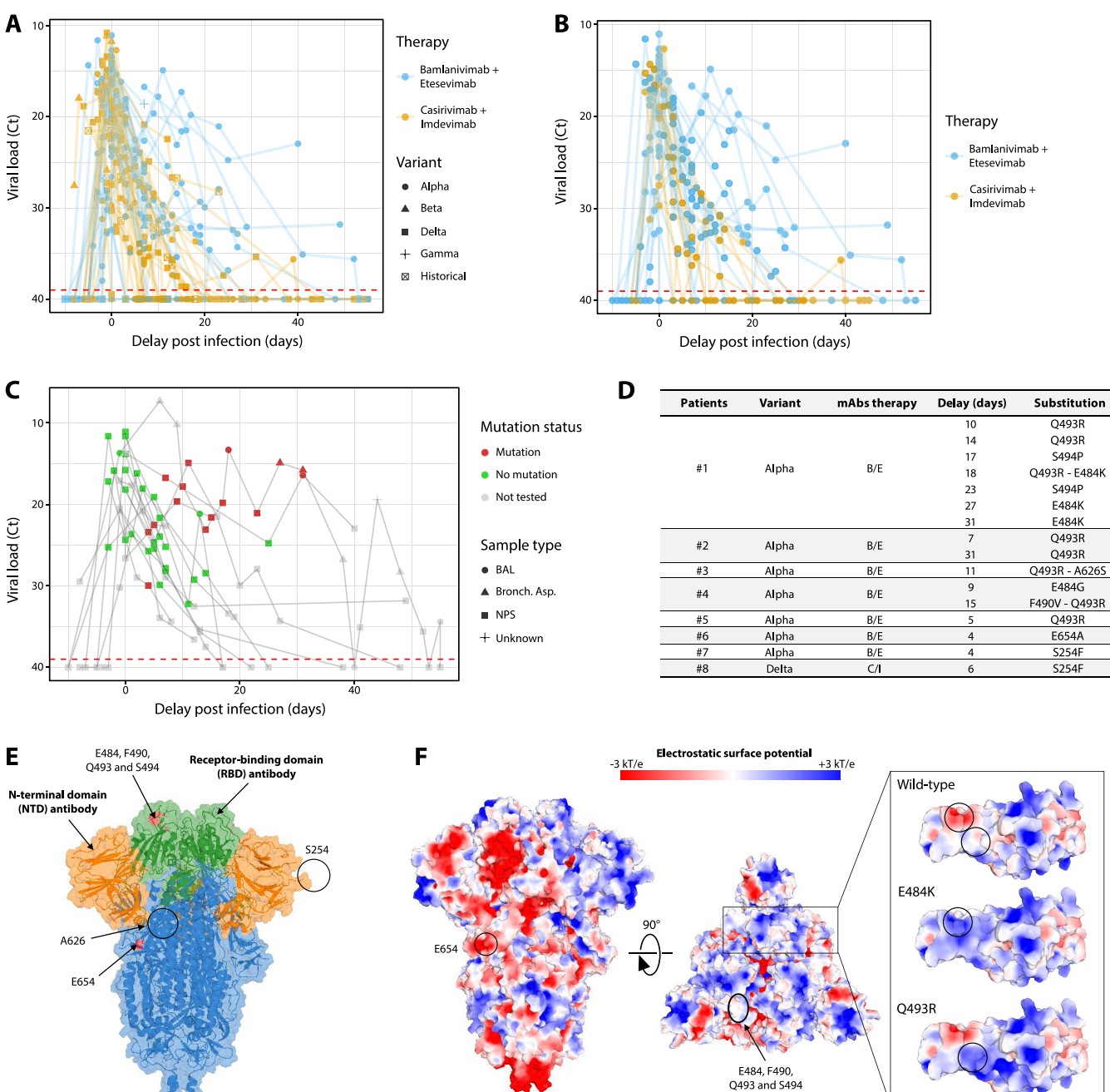

**FIG 1** Virological evolution in SARS-CoV-2 infected patients treated with MAbs dual therapy: viral load kinetics follow-up and description of viral mutation emergence and of their impact on viral spike protein. (A to C) Viral load kinetics, estimated by $C_T$ values, according to viral variants and monoclonal antibodies biotherapy (A), or Alpha variants only according to monoclonal antibodies biotherapy (B), and according to the presence of new spike mutations (indicated by the red color) and the nature of the respiratory samples (depicted by the shapes) (C). (D) Description of new mutations selected under treatment. (E) Spatial location of positions associated with selected resistance mutations upon treatment on the tertiary structure of the SARS-COV-2 Spike (PDB: 6VXX): the structure is displayed from the front views with the N-terminal domain (NTD) and receptor-binding domain (RBD) colored in orange and green, respectively. Selected mutations upon treatment are colored in red or encircled in black when the protein region is lacking in the structure. Each position associated with a mutation is labeled. The remaining Spike structure is colored in blue. (F) Electrostatic surface potential of the Spike structure, estimated with the APBS method. Electrostatic potential values are in units of kT/e at 298 K, on a scale of −3 kT/e (red) to +3 kT/e (blue). The white color indicates a neutral potential. For E484K and Q493R mutant structures, mutations were introduced individually using PyMOL (PDB: 7KMG). BAL, bronchoalveolar lavage; Bronch. Asp., bronchial aspirate; NPS, nasopharyngeal swab.

(>100- and 42-fold respectively) (7). Two other mutations selected upon treatment, F490V and S494P, are spatially close to E484K and Q493R and may disturb binding properties. The substitution of phenylalanine to valine at position 490 removed the aromatic cycle previously shown to form hydrophobic interaction with some antibodies (8), and S494P

was already associated with reduced antibody neutralization of convalescent and postimmunization sera, particularly when combined with E484K (9).

This study presents several limitations: it was conducted during the circulation of Alpha and Delta variants, and only C/I could be prescribed for Delta infections. Moreover, those two variants have been replaced to date. Future epidemiological or virological evolution of SARS-CoV-2 being hard to predict, it is still essential to better characterize MAbs therapies and mutation emergence under their selective pressure. Recently, the mutation selection was described in Delta variant infections treated by Sotrovimab (10), and ongoing studies will provide new data regarding the Omicron variant (11). Comparing selected mutations according to spike preexisting mutations across variants and MAbs therapies will help us better understand the works at play in SARS-CoV-2 antibodies neutralization escape.

## ACKNOWLEDGMENTS

This work was supported by the Agence Nationale de la Recherche sur le SIDA et les Maladies Infectieuses Emergentes (ANRS MIE), AC43 Medical Virology, and Emergen Consortium.

V.M.F. reports congress accommodation from Gilead outside the submitted work. J.M. received congress accommodation from Biostest and CSL Behring outside the submitted work. D.D. has received personal fees from Gilead-Sciences, ViiV Healthcare, MSD, and Janssen-Cilag and research grants from Gilead-Sciences and ViiV Healthcare, outside the submitted work. J.G. reports personal fees from Merck, grants and personal fees from ViiV health care, grants, and personal fees from Gilead Sciences, personal fees from Roche, personal fees from AstraZeneca, personal fees from Janssen, and personal fees from TheraThechnologies outside the submitted work. The other authors declare having no conflict of interest with the current work.

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
