## [Reviewer comments · Microbiology Spectrum]

Microbiology Spectrum

DUAL MONOCLONAL ANTIBODIES ON SARS-COV-2 ALPHA AND DELTA VARIANTS: CLINICAL AND VIROLOGICAL EFFICACY

Valentine Ferré, Nathan Peiffer-Smadja, Laura Kramer, Romain Coppée, Aïcha Kante, Margaux Debarge, Christophe Choquet, Thibault Saint Joannis, Donia Bouzid, Jonathan MESSIKA, Jennifer Le Grand, Michael Thy, Solen Kernéis, Diane Descamps, Benoit VISSEAU, and Jade Ghosn

Corresponding Author(s): Valentine Ferré, Université Paris Cité and Université Sorbonne Paris Nord, Inserm, IAME, F-75018 Paris, France; Service de Virologie, AP-HP, Hôpital Bichat - Claude Bernard, F-75018 Paris, France

Review Timeline:

Submission Date:	June 14, 2022
Editorial Decision:	August 12, 2022
Revision Received:	August 17, 2022
Accepted:	August 25, 2022

Editor: Leiliang Zhang

Reviewer(s): The reviewers have opted to remain anonymous.

Transaction Report:

DOI: <https://doi.org/10.1128/spectrum.02152-22>

August 12, 2022

Dr. Valentine Marie Ferré

Université Paris Cité and Université Sorbonne Paris Nord, Inserm, IAME, F-75018 Paris, France; Service de Virologie, AP-HP, Hôpital Bichat - Claude Bernard, F-75018 Paris, France
Paris
France

Re: Spectrum02152-22 (DUAL MONOCLONAL ANTIBODIES ON SARS-COV-2 ALPHA AND DELTA VARIANTS: CLINICAL AND VIROLOGICAL EFFICACY)

Dear Dr. Valentine Marie Ferré:

Thank you for submitting your manuscript to Microbiology Spectrum. As you will see your paper is very close to acceptance. Please modify the manuscript along the lines I have recommended. As these revisions are quite minor, I expect that you should be able to turn in the revised paper in less than 30 days, if not sooner. If your manuscript was reviewed, you will find the reviewers' comments below.

When submitting the revised version of your paper, please provide (1) point-by-point responses to the issues I raised in your cover letter, and (2) a PDF file that indicates the changes from the original submission (by highlighting or underlining the changes) as file type "Marked Up Manuscript - For Review Only". Please use this link to submit your revised manuscript. Detailed instructions on submitting your revised paper are below.

Link Not Available

Sincerely,

Leiliang Zhang

Reviewer comments:

Reviewer #1 (Comments for the Author):

This manuscript by Valentine Marie Ferré examines the efficacy of two monoclonal antibody bitherapies against CoV-2 on patients. The authors find that slower viral load decay for patients treated with the bitherapy Bamlanivimab/Etsevimab compared to the Casirivimab/Imdevimab association therapy, especially on the patients infected with an Alpha variant. By sequencing the viral genome carefully, they also find that The Q493R mutation is frequently selected after treatment of Bamlanivimab/Etsevimab on the patients infected with an Alpha variant. I only have one major comment about this mutation. The Q493R can be selected in vitro by bamlanivimab. This mutation was isolated from a patient who had coronavirus disease and was treated with Bamlanivimab/Etsevimab(PMID: 34314668). I strongly suggest the authors discuss the discovery of this mutation clearly and compare the previous study with this manuscript.

Preparing Revision Guidelines

To submit your modified manuscript, log onto the eJP submission site at <https://spectrum.msubmit.net/cgi-bin/main.plex>. Go to

Author Tasks and click the appropriate manuscript title to begin the revision process. The information that you entered when you first submitted the paper will be displayed. Please update the information as necessary. Here are a few examples of required updates that authors must address:

- point-by-point responses to the issues I raised in your cover letter
- Upload a compare copy of the manuscript (without figures) as a "Marked-Up Manuscript" file.
- Each figure must be uploaded as a separate file, and any multipanel figures must be assembled into one file.
- Manuscript: A .DOC version of the revised manuscript
- Figures: Editable, high-resolution, individual figure files are required at revision, TIFF or EPS files are preferred

Please return the manuscript within 60 days; if you cannot complete the modification within this time period, please contact me. If you do not wish to modify the manuscript and prefer to submit it to another journal, please notify me of your decision immediately so that the manuscript may be formally withdrawn from consideration by Microbiology Spectrum.

Responses to Reviewers - Spectrum02152-22R1

Reviewer #1 (Comments for the Author):

This manuscript by Valentine Marie Ferré examines the efficacy of two monoclonal antibody bitherapies against CoV-2 on patients. The authors find that slower viral load decay for patients treated with the bitherapy Bamlanivimab/Etsevimab compared to the Casirivimab/Imdevimab association therapy, especially on the patients infected with an Alpha variant. By sequencing the viral genome carefully, they also find that The Q493R mutation is frequently selected after treatment of Bamlanivimab/Etsevimab on the patients infected with an Alpha variant. I only have one major comment about this mutation. The Q493R can be selected in vitro by bamlanivimab. This mutation was isolated from a patient who had coronavirus disease and was treated with Bamlanivimab/Etsevimab(PMID: 34314668). I strongly suggest the authors discuss the discovery of this mutation clearly and compare the previous study with this manuscript.

Response to Reviewer #1:

We thank Reviewer #1 for his interest in our work. We are grateful for his comment as it is a great input for our manuscript. The Q493R mutation selection was discussed along with the addition of two references including the one suggested by the Reviewer #1 (highlighted in yellow in the revised version of the manuscript).

August 25, 2022

Dr. Valentine Marie Ferré

Université Paris Cité and Université Sorbonne Paris Nord, Inserm, IAME, F-75018 Paris, France; Service de Virologie, AP-HP, Hôpital Bichat - Claude Bernard, F-75018 Paris, France
Paris
France

Re: Spectrum02152-22R1 (DUAL MONOCLONAL ANTIBODIES ON SARS-COV-2 ALPHA AND DELTA VARIANTS: CLINICAL AND VIROLOGICAL EFFICACY)

Dear Dr. Valentine Marie Ferré:

Your manuscript has been accepted, and I am forwarding it to the ASM Journals Department for publication. You will be notified when your proofs are ready to be viewed.

Sincerely,

Leiliang Zhang
Editor, Microbiology Spectrum
